# Survival of Patients on Hemodialysis with Erectile Dysfunction

**DOI:** 10.3390/medicina56100500

**Published:** 2020-09-24

**Authors:** Radojica V. Stolic, Zoran Bukumiric, Branislav Belic, Bozidar Odalovic, Goran Relic, Sasa Sovtic, Maja Sipic, Vekoslav Mitrovic, Biljana Krdzic

**Affiliations:** 1Department of Internal Medicine, Faculty of Medical Sciences, University of Kragujevac, 34000 Kragujevac, Serbia; 2Institute of Medical Statistics and Informatics, Faculty of Medicine, University of Belgrade, 11000 Belgrade, Serbia; zoran.bukumiric@med.bg.ac.rs; 3Department of Otorhinolaryngology, Faculty of Medical Sciences, University of Kragujevac, 34000 Kragujevac, Serbia; branislav.belic21@gmail.com; 4Faculty of Medicine Pristina/K. Mitrovica, University of Pristina/K. Mitrovica, 38210 K. Mitrovica, Serbia; odalovicb@gmail.com (B.O.); dr_relicg@ptt.rs (G.R.); sovtics@yahoo.com (S.S.); drsipic@yahoo.com (M.S.); dokbiljana@gmail.com (B.K.); 5Faculty of Medicine Foca, University of East Sarajevo, 71123 Lukavica, Bosnia and Herzegovina; vekoslav_mitrovic@yahoo.com

**Keywords:** erectile dysfunction, hemodialysis, survival parameters, mortality, cardiovascular disease

## Abstract

*Background and objectives:* In patients on hemodialysis, erectile dysfunction is an independent mortality factor. This study aimed to determine the risk factors that affect the survival of hemodialysis patients with erectile dysfunction. *Materials and Methods:* During a seven-year period, erectile dysfunction was identified among the fatalities reported in patients receiving chronic hemodialysis, on the basis of the International Index of Erectile Function questionnaire. The study covered 70 patients of mean age 57 ± 6.7 years. During the examined period, 42 (60%) patients died at the mean age 57 ± 6.8 years. The study was completed by 28 (40%) patients, aged 57 ± 6.55 years. Laboratory, demographic, anthropometric, and clinical characteristics were recorded using standard methods. *Results:* Statistically significant differences between the two groups of respondents were found concerning dialysis duration (*p* < 0.001), number of leukocytes (*p* = 0.003), adequacy of hemodialysis (*p* = 0.004), intima media thickness of the carotid artery (*p* < 0.001), presence of cardiovascular disease (*p* = 0.03), residual diuresis (*p* = 0.04), and hemodiafiltration (*p* < 0.001). Hemodialysis adequacy (B = −9.634; *p* = 0.017), intima media thickness (B = 0.022; *p* = 0.003), residual diuresis (B = −0.060; *p* = 0.007), and lower rates of cardiovascular disease (B = 0.176; *p* = 0.034) were significant survival predictors among our patients with erectile dysfunction. *Conclusions:* Risk factors that are associated with improved survival of patients on hemodialysis with erectile dysfunction in our study are: preserved diuresis, high-quality hemodialysis, lower incidence of cardiovascular disease, and less thickening of the intima media of the carotid arteries.

## 1. Introduction

End-stage renal disease causes wide-ranging disturbance in men, including sexual dysfunction. Disturbances in neurovascular control, abnormal hormone levels, or psychological factors are responsible for the vast majority of erectile dysfunction that is broadly classified as psychogenic (generalized, situational), organic (vasculogenic, neurogenic, anatomic, endocrinologic) or mixed. Psychological factors include primary lack of sexual arousal, chronic disorder of sexual intimacy, depression, and performance anxiety. Physiological factors are a more common etiology and neurological disorders such as Parkinson’s disease, stroke, tumor, multiple sclerosis, and spinal cord injury are noted to be associated with erectile dysfunction. Arterial insufficiency associated with diabetes, hypertension, dyslipidemia, cigarette smoking, blunt perineal or pelvic trauma, and pelvic irradiation, tends to be the most common cause of erectile dysfunction [1,2]. The cause of erectile dysfunction in end stage renal disease is primarily organic. Sexuality is the fifth-most important life stressor in dialysis populations. Thus, erectile dysfunction is the most common disorder of sexual dysfunction impairing the quality of life of these patients [3].

Many studies have reported erectile dysfunction to be associated with macrovascular disease, but few have investigated the impact of erectile dysfunction on the progression of a microvascular disorder, such as kidney disease. Shen et al. [4] emphasized that the risk for erectile dysfunction was especially great in young patients aged < 50 years. Although the underlying mechanisms contributing to the association between erectile dysfunction and end-stage kidney disease are very complex, some possible explanations can be considered. First, metabolic comorbidities and cardiovascular disease are highly prevalent in patients with erectile dysfunction and they can worsen renal function, but they cannot explain the entire relationship. The association between erectile dysfunction and patients with end-stage renal disease may be partially due to these metabolic comorbidities, but they cannot explain the entire relationship. Second, erectile dysfunction is considered a surrogate marker of endothelial damage, which is an important pathological change. Erectile dysfunction occurs in approximately 50% of patients, before hemodialysis, and 80% during hemodialysis [5,6].

The latest findings suggest that, in the general population, men who suffer from erectile dysfunction have a 70% chance of dying early, thus, there are significant indications that erectile dysfunction is a strong predictor of premature mortality [7,8,9].

Sexual dysfunction is now considered an early manifestation of generalized cardiovascular disease and predicts cardiovascular mortality [10,11,12]. Some reports concluded that sexual dysfunction and cardiovascular disease should be regarded as two different manifestations of the same systemic disorder, while sexual dysfunction should be considered an early marker for cardiovascular disease [10].

The aim of this study was to determine the risk factors that affect the survival of hemodialysis patients with erectile dysfunction.

## 2. Material and Methods

This study is a continuation of research for which some results have already been presented [6,13]. It is organized as a non-randomized clinical study in patients treated with chronic hemodialysis in whom erectile dysfunction had been recorded. Over the seven-year period of investigation, the outcome for all patients with erectile dysfunction was recorded.

The study included 70 patients of mean age 57 ± 6.7 years. During the examined period, 42 (60%) of them died at a mean age of 57 ± 6.8 years. A total of 28 (40%) patients of mean age 57 ± 6.55 years completed the study.

Erectile dysfunction was definedon the basis of the questionnaire of the International Index of Erectile Function (IIEF)-5 [14,15], adjusted for the Serbian language. It consisted of five questions offering five responses, each one with a score. Generally, patients who had at least 21 IIEF points were defined as patients with erectile dysfunction. An IIEF score between 5–7 denotes a serious form of erectile dysfunction, from 8–11 moderate, 12–16 moderate to mild, and 17–21 a mild form, while patients who scored 22–25 points were without erectile dysfunction.

### 2.1. Laboratory, Demographic, Anthropometric and Clinical Characteristics

Biochemical analyses were done every 3 months and mean values recorded. Blood samples were obtained in Vacutainer^®^ tubes, in the middle of the week before dialysis. Plasma samples were stored at −20 °C. Analyses were made using the flow cytometric method (Beckman Coulter Inc., Fullerton, CA, USA) or spectrophotometrically on an ILAB-600 instrument (Diamond Diagnostics—USA 333 Fiske Street Holliston, MA 01746, USA) using original reagents.

Gender, duration of dialysis in months, former and active smoking, as well as alcohol consumption (at least 500 mL per week) were recorded. Body mass index (BMI) was calculated using the quotient of present body weight (kg) and the square of body height (m^2^) (kg/m^2^). On the basis of the anamnesis and medical history, the presence of cardiovascular disease (of the circulatory system, of the heart muscle, coronary heart disease, etc.) was registered. Adequacy of hemodialysis (Kt/V index), calculated by the urea kinetic model and quantified according to the formula of Daugirdas was determined for all patients [16]. In addition, the existence of residual diuresis (at least 250 mL) and the type of hemodialysis (bicarbonate/hemodiafiltration) were noted.

### 2.2. Ultrasonography of Carotid Arteries

As a parameter of atherogenesis, carotid intima-media thickness was measured using a LOGIQ P5 apparatus with camera (GE Healthcare, 9900 Innovation Drive, Wauwatosa, WI 53226, USA) and 7.5 MHz high-resolution linear-phased array color imaging transducer probes. Intima-media thickness was defined as the low-level echo grey band that did not project into the arterial lumen and was determined during the diastolic phase as the distance between the leading edge of the first and second echogenic line. Wall thickness was never determined in areas where plaque was present. Measurements were made 2 cm above and below the carotid bifurcation, with three replications on each side [17]. The average value was taken as intima-media thickness and was considered abnormal when it exceeded 0.82 mm [18]. All ultrasound examinations were done by the first author.

The study was approved by the ethics committee of the Kragujevac Clinical Center, in accordance with the Helsinki Declaration for Medical Research, ethical code KCK 03/08/15; Number 01/8104.

### 2.3. Statistical Analysis

All statistical analyses were performed in SPSS 24.0 (SPSS Inc., Chicago, IL, USA). Our results are presented as frequency, percentage, mean–standard deviation (SD) and median (where appropriate). The chi-square test, Fisher exact test and t-test were used to test the differences between groups. The significance of null-hypothesis testing was taken at the probability level *p* < 0.05.

The model of multivariate logistic regression included those predictors of a lethal outcome that were statistically significant in the model of univariate logistic regression at the probability level of 0.05 and which, based on previous studies, were known to influence the dependent variable.

## 3. Results

Table 1 shows the demographic, anthropometric, and clinical-laboratory characteristics of the respondents and differences between the group of patients with erectile dysfunction who had died and those still alive after a period of seven years. Thus, among the 70 patients of similar ages initially diagnosed with erectile dysfunction, 42 (60%) died, while 28 (40%) survived. The overall average length of dialysis treatment was 107 ± 56.1 months.

Patients who died had an average length of dialysis of 88 ± 45.5 months dialysis, while those who completed the study had been on dialysis therapy for 121 ± 56.6 months. The difference between the two groups was statistically significant (*p* < 0.001). There was also a significant difference in relation to the number of leukocytes (*p* = 0.003). The Kt/V index of the overall population tested was 1.01 ± 0.22. Among patients who died, the Kt/V index was 0.98 ± 0.18, while the value for survivors was 1.08 ± 0.25, which was significantly more favorable (*p* = 0.004). On average, intima media thickness of the carotid artery was 1 ± 0.2 mm. The value was greater for the patients who died (1.09 ± 0.18 mm) than in those who survived (0.8 ± 0.1 mm), the difference being statistically significant (*p* < 0.001). Among all respondents, 46 (66%) patients had some form of cardiovascular disease. Such disease was manifested in three quarters of the patients who died (32 or 76%) but in only half of those who survived (14 or 50%). The difference was statistically significant (*p* = 0.03). Residual diuresis was recorded in 36 (51%) subjects. Among those who died, 12 (28.6%) patients retained residual diuresis, while residual diuresis was present in relatively more of the survivors (14 or 50%; *p* = 0.04). Bicarbonate hemodialysis was given to 36 (51%) of our patients, while 34 (49%) received hemodiafiltration. In the group of patients who died, 29 (69%) were treated by bicarbonate hemodialysis but only 13 (31%) by hemodiafiltration. In contrast, among the survivors, there were 7 (25%) patients on bicarbonate dialysis and 21 (75%) on hemodiafiltration. This divergence between the two groups of patients, concerning type of hemodialysis was statistically significant (*p* < 0.001).

The univariant statistical model found that reduced Kt/V index (B = −10.301; *p* = 0.007), absence of residual diuresis (B = −3.805; *p* = 0.014), presence of cardiovascular disease (B = 1.203; *p* = 0.004) and intima media thickness (B = 17.718; *p* = 0.001) were statistically significant predictors of mortality for patients with erectile dysfunction (Table 2).

Multivariate logistic regression analysis indicated the following significant survival predictors in patients with erectile dysfunction: Kt/V index (B = −9.634; *p* = 0.017), reduced intima media thickness (B = 0.022; *p* = 0.003), presence of residual diuresis (B = −0.060; *p* = 0.007), and a reduced prevalence of cardiovascular disease (B = 0.176; *p* = 0.034) (Table 3).

## 4. Discussion

In a high-risk population for cardiovascular disease, erectile dysfunction was associated with an almost doubled risk of death. The ability of erectile dysfunction to predict cardiovascular disease seems to be age related. The association is stronger among younger men and appears to disappear or weaken markedly after the age of 60 years [19]. Other research showed a trend for reduced risk of incident cardiovascular disease in men 70 years of age or older with erectile dysfunction [20], without renal failure. These findings are interesting in the context of our study. In our sub-group of hemodialysis patients with erectile dysfunction who died during the seven-year follow-up period, we did not find significant association with years of age. The average age of both groups of respondents was 57 years, which may be one of the reasons why erectile dysfunction was not a prospective parameter for mortality. Unfortunately, in our study, we did not record the cause of death, but it is known that erectile dysfunction is a strong predictor of death from all causes. In fact, the risks of death from all causes and composite outcome increased in a stepwise manner with the progression of erectile dysfunction. The following hypothesis was confirmed, that any observed association between all-cause mortality and erectile dysfunction would be explained by the presence of an association of erectile dysfunction with cardiovascular disease death and a lack of or inconsistent association with other causes. In models adjusted for several strong confounding influences, men with erectile dysfunction have a 26% higher risk of all-cause mortality and a 43% higher risk of death due to cardiovascular disease, compared to men without erectile dysfunction [21].

While erectile dysfunction can be a distressing symptom in itself, there is increasing recognition of its importance as a risk marker for potentially life-threatening cardiovascular disease events and premature death. The results of Banks et al. [22] strongly support previous findings indicating that men with erectile dysfunction require assessment for cardiovascular disease risk, including ischemic heart disease, stroke, and peripheral vascular disease. There is a clear association of erectile dysfunction and cardiovascular diseases, while the endothelium dysfunction occurring in cardiovascular diseases may contribute to the pathogenesis of dysfunction [23]. We found an independent association between cardiovascular diseases and erectile dysfunction in the group of patients who died. This is in accordance with published results, because dialysis patients initially have a significantly greater number of comorbidities and metabolic disorders when compared to the non-dialysis population.

Other studies had strongly suggested that sexuality was already impaired during the uremic phase before dialysis, and in most patients, sexuality did not improve after the start of dialysis [24]. Also due to the atherosclerotic potential of uremic syndrome itself, we did not correlate the influence of different etiological causes of renal failure on the occurrence of erectile dysfunction. Erectile dysfunction, per se, is unlikely to be the major independent cause of cardiovascular disease, but may function as a biomarker of the severity of basic pathological processes, such as atherosclerotic and endothelial dysfunction [12]. This is in line with our results, especially in the context of the large number of leukocytes, the thick intima media of the carotid arteries, and the greater number of cardiovascular disorders in our patients who died.

Whether long duration of hemodialysis is a significant risk factor for erectile dysfunction in hemodialysis patients is still controversial, because of the relatively short period of dialysis in most study cohorts. Moreover, any difference in erectile dysfunction as a clinical risk factor between patients after short-term hemodialysis therapy and those treated long-term still remains uncertain [25,26]. We found that our patients with erectile dysfunction who died had received hemodialysis for a shorter time.

Chronic kidney disease is itself a risk factor for the development of atherosclerotic vascular disease. Vascular disease is closely associated with erectile dysfunction, so it might be expected that vascular disease and erectile dysfunction will be correlated. Such an association has been demonstrated concerning severe erectile dysfunction and carotid intima-media thickness in hemodialysis recipients [12,27,28,29,30,31]. Our patients with erectile dysfunction who died had thicker carotid arteria-media. There was a positive correlation between carotid arteria-media thickness and erectile dysfunction. Moreover, by univariate and multivariate logistic regression, we confirmed the predictive significance of this parameter on the occurrence of erectile dysfunction.

Our patients with erectile dysfunction who died had a poorer Kt/V index, when compared to those who survived. We can conclude that hemodialysis within acceptable standards for adequacy (Kt/V index 1.3) may contribute to survival of patients with erectile dysfunction. These patients have lower values for the Kt/V index than patients without erectile dysfunction [32]. Hemodialysis quality was not only lower in our patients who died but it was a parameter that showed predictive significance for survival of patients with erectile dysfunction, even though our respondents did not achieve the recommended Kt/V index. All this points to the need to establish conditions for improving the quality of hemodialysis which, certainly, would further contribute to longer survival of patients with erectile dysfunction.

The importance of residual renal function is increasingly recognized as an independent survival parameter in patients on dialysis. Some studies have concluded that patients with hemodialysis with a urine output greater than 100 mL have a reduced mortality rate and better hemodialysis [13,33,34,35]. In our case, more than 50% of the patients with erectile dysfunction who completed the study had residual diuresis. Namely, there was a significantly higher number of patients with residual urine among survivors than among the patients who died. Likewise, the univariate and multivariate logistic regression models established that survival of patients with erectile dysfunction is longer for those with preserved residual diuresis. In the available literature we found no data concerning correlation between erectile dysfunction and residual diuresis, except in our previous work where we recorded a significantly higher rate of erectile dysfunction in patients without residual diuresis [13].

### Limitations of the Study

For survival analysis, the number of our patients was not large, and the duration of the study was not very long. Therefore, our results concerning erectile dysfunction as an independent survival predictor in hemodialysis patients, should be taken with caution.

## 5. Conclusions

Risk factors that improve survival of patients on hemodialysis with erectile dysfunction in our study are: preserved diuresis, high-quality hemodialysis, lower incidence of cardiovascular disease, and less thickening of the intima media of the carotid arteries.

## Figures and Tables

**Table 1 medicina-56-00500-t001:** Basic characteristics and correlation of demographic, biochemical, and clinical characteristics of the examined patients.

	All Patients with Erectile Dysfunction (70)	Surviving Patients (28)	Patients Who Died (42)	*p*
Age, mean ± SD	57 ± 6.7	57 ± 6.55	57.5 ± 6.8	0.33
Duration of hemodialysis (months), mean ± SD	107 ± 56.1	121 ± 56.6	88 ± 45.5	<0.001 *
Leukocytes (10^9^/L), mean ± SD	6.5 ± 1.8	5.9 ± 1.68	7.3 ± 1.7	0.003 *
Erythrocytes (10^12^/L), mean ± SD	2.94 ± 0.52	2.93 ± 0.47	2.97 ± 0.54	0.15
Hemoglobin (g/L), mean ± SD	96.5 ± 13.7	90 ± 14.8	98.5 ± 12.7	0.18
Serum iron (µmol/l), mean ± SD	10.3 ± 3.8	11.2 ± 2.53	10.05 ± 4.49	0.48
Ferritin (µg/l), mean ± SD	590.4 ± 318.4	632 ± 321	582.9 ± 315.8	0.38
Transferrin saturation (%), mean ± SD	26 ± 11	28.3 ± 6.78	24.7 ± 12.9	0.27
Total calcium (mmol/L), mean ± SD	2.33 ± 0.19	2.28 ± 0.14	2.34 ± 0.21	0.14
Inorganic phosphorus (mmol/L), mean ± SD	1.65 ± 0.56	1.64 ± 0.49	1.72 ± 0.6	0.44
Parathormone (pg/mL), mean ± SD	169± 406	170 ± 406.9	153 ± 404	0.30
Creatinine (μmol/L), mean ± SD	998 ± 228.8	1023 ± 225.9	992 ± 230.5	0.37
Urea (mmol/L), mean ± SD	24.6 ± 5.5	26.5 ± 5.67	24.4 ± 5.67	0.12
Kt/V index	1.01 ± 0.22	1.08 ± 0.25	0.98 ± 0.18	0.004 *
BMI (kg/m^2^), mean ± SD	25.2 ± 5.7	24.8 ± 4.41	25.7 ± 6.46	0.45
Number of erectile dysfunction points (n)	14 ± 6.4	14.5 ± 6.56	25.7 ± 6.46	0.12
Intima-media thickness (mm), mean ± SD	1 ± 0.2	0.8 ± 0.1	1.09 ± 0.18	<0.001 *
Smokers, (n)	42	14	28	0.32
Alcohol consumption, (n)	25	9	16	0.79
Cardiovascular disease, (n)	46	14	32	0.03 *
Residual diuresis, (n)	27	15	12	0.04 *
Bicarbonate hemodialysis, (n) Hemodiafiltration, (n)	36 34	7 21	29 13	<0.001 *

SD—standard deviation; BMI – body mass index. * statistically significant differences between groups.

**Table 2 medicina-56-00500-t002:** Univariant regression model of predictive parameters of mortality for patients with erectile dysfunction.

	B	S.E.	Wald	Df	Sig.	Exp(B)	95% C.I. for EXP(B)	
							Lower	Upper
Kt/V index	−10.301	3.789	7.390	1	0.007 *	0.000	0.000	0.056
Residual dieresis	−3.805	1.554	5.991	1	0.014 *	0.022	0.001	0.469
Type of hemodialysis	−1.337	1.131	1.398	1	0.237	0.263	0.029	2.409
Cardiovascular disease	1.203	0.422	8.113	1	0.004 *	3.331	1.445	7.622
Intima media thickness	17.718	5.299	11.179	1	0.001 *	0.000	0.000	0.55
Constant	−3.157	4.776	0.437	1	0.509	0.043		

B—regression coefficient; S.E.—standard error; C.I.—confidence interval. * statistically significant parameters.

**Table 3 medicina-56-00500-t003:** Multivariate logistic regression with fatal outcome as the dependent variable.

Independent Variable	B	*p*	OR	95% C.I. for EXP(B)	
				Lower	Upper
Kt/V index	−9.634	0.017 *	<0.01	<0.01	0.18
Intima media thickness	0.022	0.003 *	1.02	1.01	1.04
Residual dieresis	−0.060	0.007 *	0.94	0.90	0.98
Type of hemodialysis	−2.153	0.209	0.12	0.00	3.34
Cardiovascular disease	0.176	0.034 *	1.19	1.01	1.40

OR—odds ratio. * statistically significant parameters.

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
