# Peer review of "Survival of Patients on Hemodialysis with Erectile Dysfunction"

_medicina, 2020, doi:10.3390/medicina56100500_

Round 1

Reviewer 1 Report

The authors aimed to assess the survival of patients on hemodialysis with erectile dysfunction. The study has some limitations. The introduction lacks of a clear and fluid explanation of the common pathways of the evaluated diseases and their possible connections. The authors stated that “The aim of our study was to determine the effect of erectile dysfunction on survival in patients on hemodialysis”, however they considered only patients with ED without a comparison with patients on hemodialysis without ED. Neither a comparison between groups divided by ED severity was done. Indeed, the conclusions “In the our study, predictive parameters of survival of patients on hemodialysis with erectile dysfunction are preserved residual diuresis, high quality hemodialysis, lower incidence of cardiovascular diseases, and reduced intima media thickness of the carotid arteria” are completely not in line with the aim. I suggest to pay more attention regarding the agreement between the purpose, the study design and the results.

Author Response

First reviewer suggestions:

I suggest to pay more attention regarding the agreement between the purpose, the study design and the results.

The target group of our study were patients with erectile dysfunction who are dialyzed by chronic hemodialysis, in order to determine the risk factors among this population (the goal is reformulated according to your recommendations) that contribute to the survival of these subjects.

Now the goal was to determine the risk factors that affect the survival on hemodialysis patients with erectile dysfunction.

Therefore, a group of patients without erectile dysfunction was not the subject of our interest.

According to your suggestions, we have rearranged the introduction.

In the first paragraph of the Introduction, lines 38 - 46, the sentences is added: Disturbances in neurovascular control, abnormal hormone levels or psychological factors are responsible for the vast majority of erectile dysfunction that is broadly classified as psychogenic (generalized, situational), organic (vasculogenic, neurogenic, anatomic, endocrinologic) or mixed. Psychological factors include primary lack of sexual arousal, chronic disorder of sexual intimacy, depression and performance anxiety. Physiological factors are a more common aetiology and neurological disorders such as Parkinson’s disease, stroke, tumour, multiple sclerosis and spinal cord injury are noted to be associated with erectile dysfunction. Arterial insufficiency associated with diabetes, hypertension, dyslipidemia, cigarette smoking, blunt perineal or pelvic trauma and pelvic irradiation, tend to be the most common cause of erectile dysfunction [1, 2].

Sentence in the line 44-45, 47-48, as well as the sentence in line 52, 57-59 are deleted.

In the Material and Methods, part of the sentence in the second paragraph, line 78, has been deleted and it now reads: Erectile dysfunction is defined done on the basis of the questionnaire of the International Index of Erectile Function (IIEF)-5 [12, 13], adjusted for the Serbian language.

According to your suggestions, we have reformulated the conclusion which now reads:

"Risk factors that improve survival patients on hemodialysis with erectile dysfunction in our study is: preserved diuresis, high-quality hemodialysis, lower incidence of cardiovascular disease and less thickening of the intima media of the carotid arteries", and now, correlates with the goal of the research.

Reviewer 2 Report

This paper reported the predictive parameters of survival of patients on hemodialysis (HD) with erectile dysfunction (HD). The importance of preserved residual dieresis, high quality hemodialysis, lower incidence of cardiovascular diseases, and reduced intima thickness of the carotid arteria. These results are considered to be acceptable, and we can recognize significance of arterial sclerosis for mortality for HD patients with ED. Accordingly, arterial sclerosis is one of the most important conditions for survival in HD patients with ED. I am sure that the authors should reveal the relationship between arterial sclerosis and the causes of disease in HD patients with ED.

Please check below points.

(1) The causes of HD in this subject.

What kind of renal dysfunctions in this subject? DM? Glomerulonephritis? The severity of arterial sclerosis should be difference depending on real dysfunction type. If the authors have these data, please describe them. If not, add the authors’ comment about this issue.

(2) The causes of ED in this subject.

What were the causes of ED? Functional? Mechanical?

In functional ED patients, arterial sclerosis might not be so severe. .If the authors have these data, please describe them. If not, add the authors’ comment about this matter.

(3)  The causes of the death in this subject.

The causes of the death in this subject are important. Did they correlate to cardiovascular disease? If the authors have these data, please describe them. If not, add the authors’ comment about this problem.

Author Response

(1) The causes of HD in this subject.

What kind of renal dysfunctions in this subject? DM? Glomerulonephritis? The severity of arterial sclerosis should be difference depending on real dysfunction type. If the authors have these data, please describe them. If not, add the authors’ comment about this issue.

Due to the atherosclerotic potential of uremic syndrome itself, we did not correlate the influence of different etiological causes of renal failure on the occurrence of erectile dysfunction. Because they are studies had strongly suggested that sexuality was already impaired during the uremic phase before dialysis, and in most patients, sexuality did not improve after the start of dialysis.

At the beginning of the third paragraph, in the Discussion section, the sentence is inserted: Because they are studies had strongly suggested that sexuality was already impaired during the uremic phase before dialysis, and in most patients, sexuality did not improve after the start of dialysis [24], and, on the other hand, due to the atherosclerotic potential of uremic syndrome itself, we did not correlate the influence of different etiological causes of renal failure on the occurrence of erectile dysfunction.

(2) The causes of ED in this subject.

What were the causes of ED? Functional? Mechanical?

In functional ED patients, arterial sclerosis might not be so severe. If the authors have these data, please describe them. If not, add the authors’ comment about this matter.

The second sentence, the first paragraph in the Introduction: Disturbances in neurovascular control, abnormal hormone levels or psychological factors are responsible for the vast majority of erectile dysfunction that is broadly classified as psychogenic (generalized, situational), organic (vasculogenic, neurogenic, anatomic, endocrinologic) or mixed. Psychological factors include primary lack of sexual arousal, chronic disorder of sexual intimacy, depression and performance anxiety. Physiological factors are a more common aetiology and neurological disorders such as Parkinson’s disease, stroke, tumour, multiple sclerosis and spinal cord injury are noted to be associated with erectile dysfunction. Arterial insufficiency associated with diabetes, hypertension, dyslipidemia, cigarette smoking, blunt perineal or pelvic trauma and pelvic irradiation, tend to be the most common cause of erectile dysfunction.

(3) The causes of the death in this subject.

The causes of the death in this subject are important. Did they correlate to cardiovascular disease? If the authors have these data, please describe them. If not, add the authors’ comment about this problem.

The last two sentences in the first paragraph in the Discussion section:

Unfartunately, we did not record the cause of death, but it is known that it is erectile dysfunction is a strong predictor of death from all causes.

In fact, the risks of death from all causes and composite outcome increased in a stepwise manner with the progression of erectile disfunction.

The hypothesis that any observed association between all-cause mortality and erectile dysfunction would be explained by the presence of an association of erectile dysfunction with cardiovascular disease death and a lack of or inconsistent association with other causes was confirmed. In models adjusted for several strong confounding influences, men with erectile dysfunction have a 26% higher risk of all-cause mortality and a 43% higher risk of death due to cardiovascular disease, compared to men without erectile dysfunction.

Reviewer 3 Report

This timely manuscript focuses on a very interesting topic. I would congratulate with the Authors for the completeness of the methods and result section. It is very well written. Methods are in line with the study purpose, and are exhaustively written. Statistics is well explained. Results are presented in a clear manner.

 I would only suggest the following comment:

Are total testosterone serum level available? May this variable impact on survival?

Author Response

Response to the third reviewer's suggestion:

Are total testosterone serum level available? May this variable impact on survival?

Unfortunately, we were not able to do the level of testosterone, which is known to have a significant impact on the occurrence of erectile dysfunction.

Round 2

Reviewer 1 Report

After the revision the text is ready for publication